# Application of Electronic Health Record Text Mining: Real-World Tolerability, Safety, and Efficacy of Adjuvant Melanoma Treatments

**DOI:** 10.3390/cancers14215426

**Published:** 2022-11-03

**Authors:** Sylvia A. van Laar, Ellen Kapiteijn, Kim B. Gombert-Handoko, Henk-Jan Guchelaar, Juliette Zwaveling

**Affiliations:** 1Department of Clinical Pharmacy & Toxicology, Leiden University Medical Center, 2333 ZA Leiden, The Netherlands; 2Department of Medical Oncology, Leiden University Medical Center, 2333 ZA Leiden, The Netherlands

**Keywords:** adjuvant treatment, melanoma, immune-checkpoint inhibitors, targeted therapy, real-world data, electronic health record, text mining

## Abstract

**Simple Summary:**

Recently, nivolumab, pembrolizumab, both immune-checkpoint inhibitors (ICIs) and the combination of dabrafenib plus trametinib (D + T) were registered as adjuvant melanoma treatments, to prevent recurrence. The aim of this paper was to retrospectively review the benefits and risks of these treatments in clinical practice, by extracting data from electronic health records with a text-mining tool. In a population of 122 patients, 55 used nivolumab, 48 used pembrolizumab and 20 used D + T, and we found that the ICIs were better tolerated than D + T. However, the frequent adverse events of D + T are reversible and include pyrexia and fatigue. ICIs show immune-related chronic adverse events, and chronic thyroid-related adverse events occurred frequently. The efficacy results, including the recurrence-free survival, are promising; however, the follow-up was too short for conclusions. This study furthermore showed that the application of text-mining is a valuable method to collect data for the evaluation of adjuvant melanoma treatments.

**Abstract:**

*Introduction***:** Nivolumab (N), pembrolizumab (P), and dabrafenib plus trametinib (D + T) have been registered as adjuvant treatments for resected stage III and IV melanoma since 2018. Electronic health records (EHRs) are a real-world data source that can be used to review treatments in clinical practice. In this study, we applied EHR text-mining software to evaluate the real-world tolerability, safety, and efficacy of adjuvant melanoma treatments. *Methods*: Adult melanoma patients receiving adjuvant treatment between January 2019 and October 2021 at the Leiden University Medical Center, the Netherlands, were included. CTcue text-mining software (v3.1.0, CTcue B.V., Amsterdam, The Netherlands) was used to construct rule-based queries and perform context analysis for patient inclusion and data collection from structured and unstructured EHR data. *Results:* In total, 122 patients were included: 54 patients treated with nivolumab, 48 with pembrolizumab, and 20 with D + T. Significantly more patients discontinued treatment due to toxicity on D + T (N: 16%, P: 6%, D + T: 40%), and *X*^2^ (6, *n =* 122) = 14.6 and *p* = 0.024. Immune checkpoint inhibitors (ICIs) mainly showed immune-related treatment-limiting adverse events (AEs), and chronic thyroid-related AE occurred frequently (hyperthyroidism: N: 15%, P: 13%, hypothyroidism: N: 20%, P: 19%). Treatment-limiting toxicity from D + T was primarily a combination of reversible AEs, including pyrexia and fatigue. The 1-year recurrence-free survival was 70.3% after nivolumab, 72.4% after pembrolizumab, and 83.0% after D + T. *Conclusions:* Text-mining EHR is a valuable method to collect real-world data to evaluate adjuvant melanoma treatments. ICIs were better tolerated than D + T, in line with RCT results. For BRAF+ patients, physicians must weigh the higher risk of reversible treatment-limiting AEs of D + T against the risk of long-term immune-related AEs.

## 1. Introduction

In 2020, approximately 300,000 patients worldwide were diagnosed with melanoma of the skin, accounting for 1.7% of all cancer diagnoses [1]. In the Netherlands, the incidence of melanoma has more than doubled in the last 20 years from 18/100,000 people in 2001 to 43/100,000 in 2021 with a mortality rate of 4–5/100,000 [2].

The introduction of immune checkpoint inhibitors (ICIs) (e.g., nivolumab, pembrolizumab and ipilimumab) and inhibitors of the mitogen-activated protein kinase pathway (e.g., BRAF inhibitors and MEK inhibitors) has improved the treatment of metastatic melanoma in the past years [3]. In 2018, nivolumab (N), pembrolizumab (P), and the combination of dabrafenib plus trametinib (D + T) were registered by the European Medicines Agency as adjuvant treatment in resected stage III and IV of melanoma [4,5,6,7]. Patients receive adjuvant treatment after surgical resection, for a maximum of 12 months or until treatment-limiting toxicity or recurrence of the disease.

The results of the phase III trials that were the basis for the indication expansion are summarized in Table 1. Eligibility criteria differed between trials, e.g., the inclusion of resected stage IV in the CheckMate 238 trial, and inclusion only of patients with a BRAF V600E or V600K mutation in the COMBI-AD trial [8,9]. Furthermore, all treatments were superior to their comparator regarding recurrence-free survival but concluding overall survival results are currently unknown.

As these patients are, in principle, cured after surgery, the safety profile of adjuvant therapy may be even more relevant than in the setting of palliative treatment. Regarding safety, nivolumab and pembrolizumab are comparable, but differ from dabrafenib plus trametinib treatment. Grade 3 or 4 adverse events (AEs) occurred in 25.4% and 31.6% of the study populations of nivolumab and pembrolizumab versus 41% in dabrafenib-plus-trametinib-treated patients, and 9.7% and 13.8% of the patients discontinued treatment due to toxicity, compared to 26% in the dabrafenib and trametinib treated. However, most reported AEs on ICI were skin reactions and fatigue. All immune-related (ir)AEs combined also have a high incidence, of which hypothyroidism and hyperthyroidism—both manifestations of thyroiditis—were most frequent, but also include, e.g., diabetes mellitus type I [8,12]. Of these immune-related endocrine toxicities, it is presumed they result in permanent and irreversible dysfunction, resulting in lifelong hormone supplementation [14]. This contrasts with the AEs of dabrafenib plus trametinib, of which pyrexia, fatigue and nausea are most common and easily reversible after treatment interruption [9,15].

Since RCT results may not represent the benefits and risks of treatments in clinical practice, real-world data can add insightful information and support decision making [16]. The electronic health record (EHR) is one of the sources that contains relevant real-world data for cancer treatment evaluation, since it includes, e.g., hospital visits, patient demographics, medication orders, laboratory data, vital signs, and imaging results [17]. However, as most data are captured in free-text notes, manual chart review is still the standard method for data extraction from EHR, which is very labor intensive and time consuming. Previously, we validated a text-mining tool to extract data from EHR for the evaluation of metastatic renal cell carcinoma treatments (mRCCs) and showed that this method is accurate and seven times faster than manual data extraction [18]. This tool has already been used in other real-world studies, e.g., to review mRCC treatment patterns and outcomes in two hospitals, to review the use of granulocyte-colony stimulating factor and incidence of febrile neutropenia in breast cancer patients [19,20].

The aim of this study was to apply a text-mining tool to retrospectively review the tolerability, safety, and efficacy of new adjuvant treatments for melanoma in a Dutch clinical hospital setting.

## 2. Methods

We performed a retrospective EHR review by collecting data from EHR through text-mining. The study protocol was reviewed and approved by the Medical Ethics Review Committee of the Leiden University Medical Center (LUMC), Leiden, the Netherlands, which waived the need for informed consent.

### 2.1. Electronic Health Record Text Mining

We retrospectively identified patients and collected all data from the electronic health record with text-mining software (CTcue B.V., Amsterdam, the Netherlands). This software program enables rule-based text-mining of the EHR, which can extract both data from structured data (e.g., medical prescriptions and laboratory values) and unstructured, free-text notes (e.g., medical notes and correspondence). Previously, we validated this tool and method to collected real-world data of renal cell carcinoma treatments in clinical practice [18].

Figure 1 shows the four steps taken within the program, per tab, from start until the export of the final dataset. First, rule-based queries were constructed in the criteria tab, to select the correct patient population. Secondly, the cohort tab gives a brief overview of the selected patients and the criteria on which they were included. The context of a datapoint can be further expanded by selecting the datapoint; in case of free-text notes, the system shows the complete form or report with the specified key word (combinations). An overview of all cases that matched the search results, e.g., all free-text forms with a hit, added to all structured data points, e.g., lab values between a specific period, is provided. The patient cohort is confirmed by reviewing all hits in unstructured text. In case of missing data necessary for patient inclusion, a patient is not included. Similarly, rule-based queries were constructed for the collection of data in the questions tab and all data from unstructured text were confirmed per patient by context analysis.

### 2.2. Patient Population

Patients were included in a university medical center, the LUMC, Leiden, the Netherlands. We aimed to include all patients aged 18 years and older with melanoma if they started adjuvant treatment of nivolumab or pembrolizumab, and dabrafenib plus trametinib between January 2019 and October 2021. Therefore, the search criteria included: 1. Patients who received nivolumab, pembrolizumab or dabrafenib plus trametinib; 2. Patients with a reimbursement code (diagnosis treatment code) specific for melanoma; 3. A free-text note that confirmed use of the treatment (e.g., search for terms as “nivolumab”, “nivo”, “Opdivo^®^”); 4. A free-text note that confirmed the adjuvant aspect of the treatment (e.g., “stage III”, “adjuvant”). The complete list of criteria is available in Appendix A.

### 2.3. Data Collection

The following patient characteristics were collected at the start of treatment: age, sex, performance status, disease stage (according to the 7th edition of the American Joint Committee on Cancer (AJCC)), primary tumor location, subtype, brain metastases, lactate dehydrogenase (LDH), ulceration status, and BRAF-, NRAS- and KIT mutation status. Moreover, the following outcomes were collected: the time on treatment, the reason to end the treatment, most common AEs as reported in the RCTs (abdominal pain, arthralgia, asthenia, chills, cough, diarrhea, dyspnea, fatigue, headache, nausea, pruritus, pyrexia, and rash, with additional irAEs (colitis, diabetes mellitus type 1, hepatitis, hyperthyroidism, hypothyroidism, pneumonitis)), all treatment-limiting AEs, and recurrence of melanoma. The common terminology criteria for adverse events (CTCAEs) v5.0 were used to grade the severity of adverse events in the context analysis.

### 2.4. Statistical Analysis

Data management and analysis were performed using R Statistical Software (v4.1.1; R CoreTeam 2021, Vienna, Austria). We used descriptive statistics to describe the patient and disease characteristics. The time on treatment and time until recurrence per patient were visualized in a swimmer plot and the median time on treatment per reason to end treatment was calculated. Chi-square test was performed to test for differences in tolerability (treatment-limiting toxicity) between treatments. Per treatment, the RFS was analyzed with the Kaplan–Meier method and visualized in a survival plot. All statistical analyses were exploratory.

## 3. Results

By text-mining, 217 patients with melanoma were identified in the EHR. Patients who received their systemic treatment as palliative treatment (*n =* 93) were excluded after context analysis (Figure 2).

In total, 122 melanoma patients were included in the study, 54 patients started with nivolumab, 48 with pembrolizumab and 20 patients with dabrafenib plus trametinib between January 2019 and October 2021. The median age of the patients was 59 years (range: 21–84), and more than half of the patients were male (61.5%). All baseline patient and disease characteristics are shown in Table 2.

### 3.1. Time on Treatment

The adjuvant treatment was ended for 97 (79%) of the patients, and 25 patients were still on treatment at the time of data extraction (Figure 3). The mean time on treatment of patients, who ended treatments was 10.2 months for nivolumab (interquartile range (IQR): 6.4–12.0), 11 months for pembrolizumab (IQR: 4.2–12.4), and 8.4 months for dabrafenib plus trametinib (IQR: 2.2–11.6).

### 3.2. Adverse Events

The reported AEs are summarized in Table 3. Fatigue was the most reported AE in all groups (N: 77.8%, P: 75.0%, D + T: 80.0%), followed by diarrhea (44.5%) and headache (29.6%) for patients treated with nivolumab; diarrhea (23.0%) and nausea, pruritus, and hypothyroidism (all 18.8%) for patients treated with pembrolizumab; and pyrexia (65.0%) and nausea (55.0%) for dabrafenib plus trametinib treatment. The adverse events include the following severe AEs: hepatitis (3.7%), asthenia (1.9%) and diarrhea (1.9%) on nivolumab, diarrhea (4.2%) and colitis (2.1%) on pembrolizumab, and nausea, incl. vomiting (5.0%) on dabrafenib plus trametinib.

### 3.3. Treatment-Limiting Toxicity

In total, in twenty (16.4%) patients, treatment was ended early due to toxicity. The reason to end the adjuvant treatment due to toxicity was significantly different between the treatment groups, with more patients ending treatment in the dabrafenib plus trametinib group (8/15), than patients treated with nivolumab (9/46), and pembrolizumab (3/36), *X*^2^ (6, *n =* 122) = 14.6, *p* = 0.024. Eight patients stopped within the first three months (N: 2, P: 1, D + T: 5) (Figure 3), and in this group, in total, 28 treatment-limiting AEs were reported (Table 4). IrAEs were most reported as treatment-limiting in patients treated with nivolumab (8/9) and pembrolizumab (2/3), including hepatitis, colitis, pneumonitis, and thyroiditis. Pyrexia (5/8), skin disorders (4/8), chills and nausea (both 3/8) were most reported in patients treated with dabrafenib plus trametinib (*n* = 8). An overview of treatment-limiting AEs per patient is available in Appendix A.

### 3.4. Recurrence

In total, 28 patients (23.0%) ended treatment due to recurrence, 15 of the patients were treated with nivolumab, 12 with pembrolizumab and 1 with dabrafenib plus trametinib (Figure 3). The recurrence probability at 1 year was 70.3% (95% CI: 58.1–85.0) for nivolumab, 72.4% (95% CI: 60–87.3) for pembrolizumab, and 83.0% (95% CI: 67.1–1) for dabrafenib plus trametinib (Figure 4). No median RFS was reached for any of these treatments within the maximum follow-up period of 13 months.

## 4. Discussion

In an effort to optimize data extraction from EHRs and to evaluate the tolerability, safety and efficacy of nivolumab, pembrolizumab and dabrafenib plus trametinib as adjuvant treatments for resected stage III and IV melanoma, we performed a retrospective study by text-mining the EHR in a university hospital. This is the first real-world study on adjuvant treatment in resected stage III and IV melanoma with retrieval of clinical data by text-mining. By collecting a sufficient amount of data, this study shows that text-mining EHR is a valuable and effective new method. The majority of the 122 included patients received ICI (N: 45%, P: 39%), and 16% of our patients received dabrafenib plus trametinib, and we found that adjuvant ICI was better tolerated than dabrafenib plus trametinib, which had a higher risk of treatment-limiting AEs, although reversible.

Even though included patients were treated in the adjuvant setting for melanoma, patient populations slightly differ since the studied treatments are applied for specific indications within adjuvant treatment (e.g., D + T for BRAF positive patients only, and nivolumab also for resected stage IV). Furthermore, dabrafenib plus trametinib was only available through an expanded access program until November 2020, potentially influencing treatment choice. Therefore, our results should be interpreted with caution when comparing treatments head-to-head.

### 4.1. Tolerability

As the investigated treatments are administered adjuvant after surgery, the accepted toxicity, and therefore tolerability, might be valued differently than during palliative treatment. In this small study, significantly more patients treated with dabrafenib plus trametinib ended their treatment due to treatment-limiting toxicity (53%), than after treatment with ICI (N: 20%, P: 8%). This is not completely unexpected, as the rate for treatment-limiting toxicity in COMBI-AD trial of dabrafenib plus trametinib was also higher (26%), than in the CheckMate 238-trial for nivolumab (8%), and the EORTC 1325/keynote-045 trial for pembrolizumab (14%) [8,9,12]. However, both for dabrafenib plus trametinib and nivolumab, the relative number of patients who ended treatment due to toxicity was higher than in the trials. This was also observed in the studies of de Meza et al. and Hoffmann et al., which showed higher treatment-limiting AEs on ICI in clinical practice [21,22]. As far as we know, no other real-world data have been published with treatment-limiting toxicity rates for dabrafenib plus trametinib.

### 4.2. Safety

#### 4.2.1. Immune Checkpoint Inhibitors

In this study, not all severe (≥grade 3) AEs were treatment-limiting, and vice versa. However, the incidence of one or two concurring irAEs was mostly treatment-limiting. IrAEs are characterized by auto-reactive T-cells, and can affect many organs, as was demonstrated in this study by the variety of treatment-limiting AEs [23,24,25]. Furthermore, we reported severe hepatitis (*n =* 2), asthenia (*n =* 1), and diarrhea (*n =* 1) during treatment with nivolumab and diarrhea (*n =* 2), and colitis (*n =* 1) during pembrolizumab. De Meza et al. did report higher rates of severe AEs in clinical practice compared to the RCTs [21]. Due to the low incidence of the individual severe AEs, we could not determine if this was the case for our study population.

Besides, manifestations of thyroiditis, hyperthyroidism (N: 15%, P: 13%) and hypothyroidism (N: 20%, P: 19%) seemed to occur more often in this study than in the trials (hyperthyroidism, N: 8%, P: 10%; hypothyroidism: N: 10%, 14%) [8,12]. Even though a thyroid-related AE does not necessarily lead to treatment termination, patients often need lifelong hormone replacement [26], and therefore the occurrence of this type of AE might have a severe impact on an individual patient’s quality of life.

The chronic aspect and the recurrence of irAEs can play a role in the treatment decision. In our study, we did not determine the duration of AEs, but Patrinely et al. showed that in clinical practice irAEs appeared more often than was shown in clinical trials and were frequently persistent [27]. Furthermore, restarting therapy after a significant irAE was shown to result in another irAE in 50% of the patients in both a population with renal cell carcinoma and lung cancer, even though this was mostly with a lower severity rate [28,29].

#### 4.2.2. Dabrafenib Plus Trametinib

For dabrafenib plus trametinib treatment, the only reported severe AE was nausea, even though a total of 28 AEs contributed to treatment discontinuation in eight patients. In contrast to ICI, it was always combinations of AEs (two up to six) that were treatment-limiting. Pyrexia (5/8) was the most frequently reported, which is not surprising, as pyrexia specifically is known to occur during treatment with dabrafenib plus trametinib [30]. In three occurrences, it was reported with, at least, chills and nausea. In total, pyrexia occurred in total in 13/20 patients, comparable to the RCT [9]. Additionally, even though in the RCT pyrexia was also identified as the AE most often (9%) resulting in treatment discontinuation, [31], the rate of five out of twenty patients in our population seemed higher.

In this study, five of the eight patients on dabrafenib plus trametinib with treatment-limiting toxicity ended treatment within three months. Early treatment-limiting toxicity (≤3 months) is especially undesirable since these patients might not benefit at all from adjuvant treatment. Remarkably, four out of the five patients were female. However, as these patient numbers are small, we were not able to substantiate this with statistical analysis. Atkinson et al. showed that the incidence of AEs on dabrafenib plus trametinib is the highest during the first three months [31], which is in accordance with the pattern we observed. To encourage patients to remain on treatment, patient education and empowerment could be useful. Mansfield et al. showed that the maximum acceptable risk for pyrexia in melanoma patients receiving dabrafenib plus trametinib was higher when the awareness of the benefits of the therapy is higher. Therefore longer follow-up results on efficacy of the COMBI-AD trial might be beneficial [32]. Furthermore, the COMBI-APlus trial showed that treatment interruption of both dabrafenib and trametinib in case of pyrexia at a temperature of ≥38 °C, instead of discontinuation of only dabrafenib in case of pyrexia of ≥38.5 °C, helped patients in the long term to remain on treatment [31].

### 4.3. Efficacy

The preliminary 1-year recurrence-free survival rates overlapped with the results of the RCTs [8,9,12]. Even though this is indicative for comparable efficacy, and RFS seems to be a valid surrogate endpoint for overall survival for adjuvant melanoma therapy, it is too soon for conclusions on real-world effectiveness [33]. De Meza et al. showed comparable 1-year survival rates of 87.0% for the stage IIIA, and 76.5% for stage IIIB, and 60.3% for stage IIIC in a population treated with ICI in Dutch clinical practice [21]. Hoffmann et al. showed a RFS at 1-year of 77.1% after nivolumab and 63.5% after pembrolizumab in a Swiss population [22]. Furthermore, even though Koelblinger et al. showed a 1-year RFS rate of 64.8% in an Austrian population, they showed comparable distant metastasis free-survival compared to the Checkmate-238 trial (77.4% vs. 80%), and concluded treatment cost-effectiveness for the Austrian population [34]. All real-world studies showed comparable effectiveness to the RCTs, which is in line with our premature results. Hoffmann et al. is to our knowledge the only published study reviewing dabrafenib plus trametinib in clinical practice; however, they only included three patients who finished their treatment [22].

### 4.4. Eligibility Criteria

A Dutch nationwide registry showed that 40% of the patients who received treatment for advanced melanoma were ineligible for phase III trials [35]. However, our study population with patients who received adjuvant treatment, in general, met the criteria from the trials. Key exclusion criteria in the phase III trials for adjuvant treatment were ECOG performance status ≥ 1, ocular (N, D + T), uveal (N) or mucosal (D + T) melanoma, previous systemic therapy for melanoma (all), auto-immune disease (N, P), uncontrolled infection (P) or use of systemic glucocorticoids (N, P) [8,9,12]. Even though the performance status is unknown for 35% of our patients—comparable to other real-world studies [36,37]—there was no reason to believe this would be significantly different from phase III trials. Only seven (5.7%) patients that had previous systemic therapy for melanoma did not meet the eligibility criteria, including neo-adjuvant treatment in the PRADO trial [38] and patients who switched adjuvant treatments after early AEs.

### 4.5. Need for Real-World Data

None of the treatments in this study, and in general, is clearly superior to the others regarding the tolerability, safety and efficacy combined, and proving superiority was not the aim of this study. However, for patients with a BRAF-mutation, a choice between adjuvant treatment with ICI or dabrafenib plus trametinib must be made. Overall, this study reflects the safety profiles shown in the RCTs; however, incidence of thyroid-related irAEs during ICI, often chronic, and treatment-limiting AEs during both nivolumab and dabrafenib plus trametinib seemed higher. Furthermore, effectiveness, estimated by the 1-year RFS, was comparable with the trials. Weilandt et al. showed that, in a discrete choice experiment, overall response rate is the most important parameter for patients when choosing a treatment, followed by the 2-year survival rate, type of adverse events and the probability of treatment-limiting AEs. However, individual (e.g., age, sex, partnership) and disease-related (e.g., tumor burden, experience with ICI or BRAF-MEK inhibitors) resulted in variation of the preferences [39]. Livingstone et al. showed that these values overlap with factors Australian physicians and nurses consider in the recommendation of ICI to patients [40].

To be able to inform melanoma patients of the risks and benefits related to their personal characteristics, more data are needed, which underlines the need for real-world data [41]. This study shows that these data can be collected with text-mining from EHR. Although this study represents only the population of a single university hospital, this is one of the fourteen melanoma centers in the Netherlands. Data on this population have also been collected from EHR by data managers for the Dutch Melanoma Treatment Registry since 2011. However, this is done manually by data managers, is updated regularly and does not include <grade 3 adverse events [21]. This study shows that data extraction using text mining is feasible, and in the future, can potentially be extended to other hospitals and registries, for efficient inclusion of larger populations.

### 4.6. Strengths and Limitations

We used rule-based text-mining to detect patients, and collect characteristics, outcomes, and adverse events from electronic health records. The advantage of this method is the faster and more standardized data extraction. Furthermore, the set of queries can repeatedly be reused to review the status of adjuvant melanoma treatments in this hospital in the future, for example, on a yearly basis, and has the potential to be implemented in other hospitals treating melanoma patients.

However, due to its retrospective design, the extracted data are both limited by the information stored in the EHR, which is prone to have missing data [42], and by the terms included in the queries [43]. In this study, we aimed for a high sensitivity, by using extensive lists of keywords for the searches in unstructured text, and additionally performed visual context analysis to confirm outcomes, resulting in limited missing data. However, certain information can be underreported.

## 5. Conclusions

This was the first real-world study on adjuvant treatment in resected stage III and IV melanoma with the retrieval of clinical data by text-mining. In this study, we show that text-mining EHR data are a valuable method to evaluate the tolerability, safety, and efficacy of adjuvant melanoma treatments. In this population, we found a higher tolerability of ICI, as compared to dabrafenib plus trametinib, primarily due to pyrexia and fatigue. Regarding safety, this study shows results comparable to the clinical trial data, except for a higher incidence of treatment-limiting AEs for both nivolumab and dabrafenib plus trametinib, thyroid-related irAEs for nivolumab and pembrolizumab, and some mild adverse events. Furthermore, all 1-year RFS rates were comparable. Implementation of text-mining of EHR in multiple hospitals can further improve the efficiency of capturing real-world data.

## Figures and Tables

**Figure 1 cancers-14-05426-f001:**
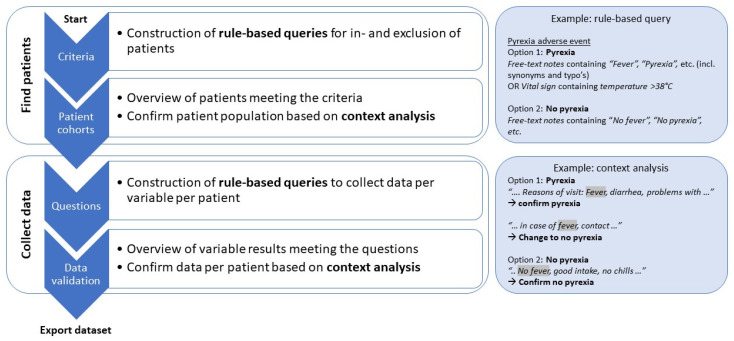
Steps taken to find patients and collect data with the text-mining tool, including the construction of rule-based queries and context analysis.

**Figure 2 cancers-14-05426-f002:**
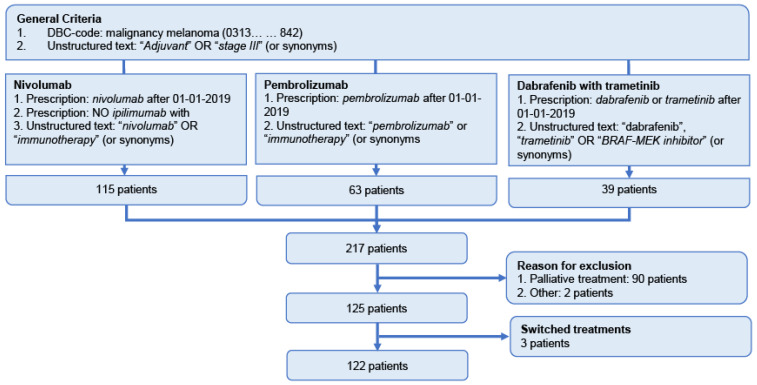
Patient flow chart. DBC-code: diagnosis-treatment combination code.

**Figure 3 cancers-14-05426-f003:**
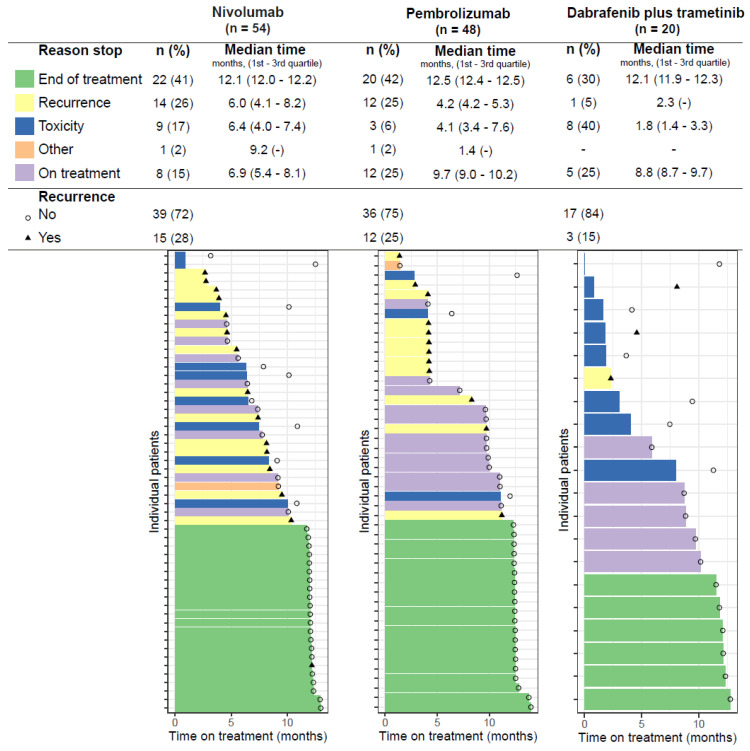
Patients’ time on treatment, causes of treatment discontinuation, and recurrence per treatment.

**Figure 4 cancers-14-05426-f004:**
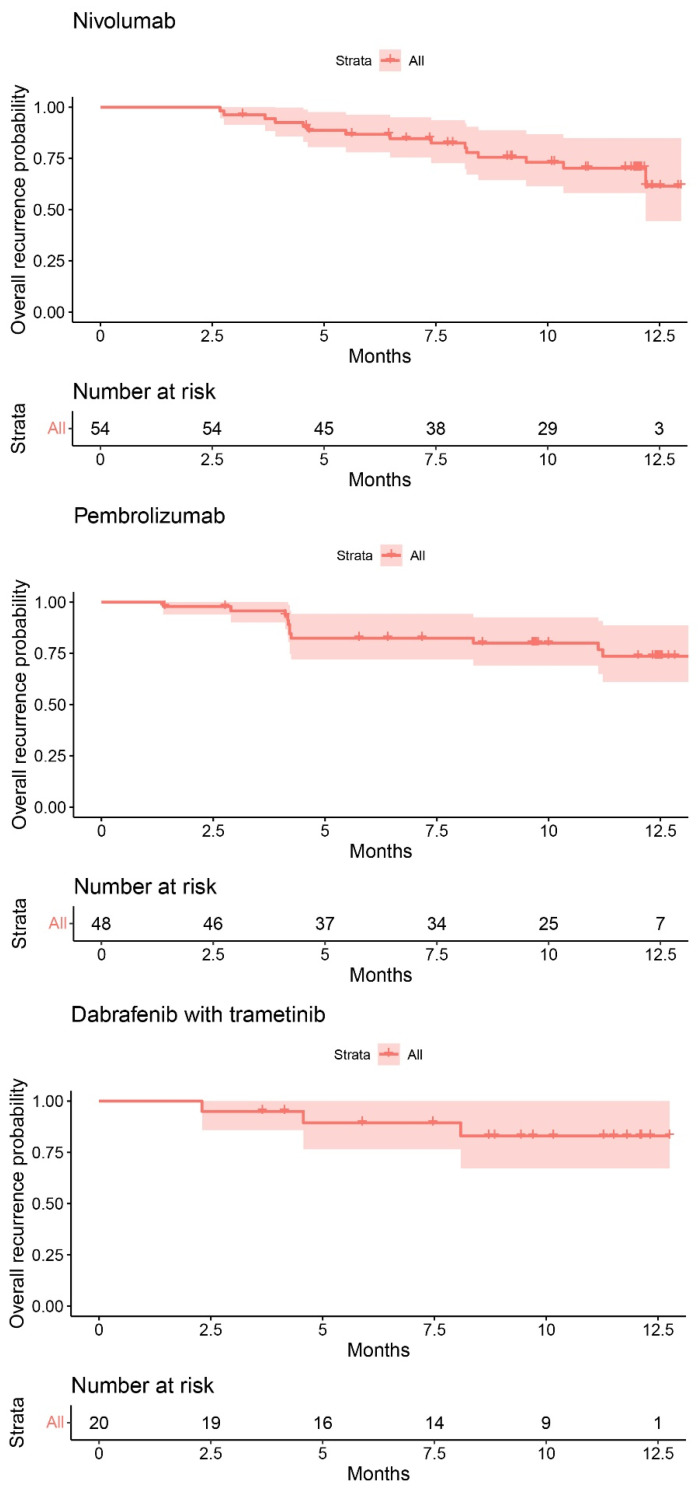
Kaplan–Meier curves representing the recurrence-free survival (RFS) of nivolumab, pembrolizumab, and dabrafenib plus trametinib 1-year RFS was 70.3% (95% confidence interval (CI): 58.1–85.0) for nivolumab, 72.4% (95% CI: 60–87.3) for pembrolizumab, and 83.0% (95% CI: 67.1–100) for dabrafenib plus trametinib.

**Table 1 cancers-14-05426-t001:** Phase III trial results for adjuvant treatments of melanoma.

	Nivolumab	Pembrolizumab	Dabrafenib Plus Trametinib
Phase III trial	Checkmate 238 [8,10]	EORTC/KEYNOTE-054 [11,12]	COMBI-AD [9,13]
Comparator	Ipilimumab	Placebo	Placebo
Eligibility criteria	Resected stage IIIb, IIIc and IV melanoma, ECOG PS: 0 or 1	Resected stage IIIa, IIIb, and IIIc melanoma, ECOG PS: 0 or 1	Resected stage IIIa, IIIb, or IIIc melanoma with a BRAF V600E or V600K mutation, ECOG PS: 0 or 1.
Recurrence-free survival	1-year	1-year	1-year
	NIV: 71%	PEM: 75%	D + T: 88%
	IPI: 61%	PLA: 61%	PLA: 56%
	4-year	3.5-year	5-year
	NIV: 51.7%	PEM: 60%	D + T: 52%
	IPI: 41.2%	PLA: 41%	PLA: 36%
	HR: 0.71	HR: 0.59	HR: 0.51
	95% CI = 0.60–0.86	95% CI = 0.49–0.70	95% CI = 0.42–0.61
Overall survival	4-year	-	3-year
	NIV: 77.9%		D + T: 86%
	IPI: 76.6%		PLA: 77%
	HR: 0.87		HR: 0.57
	95% CI = 0.66–1.14		95% CI = 0.42–0.79 *
≥Grade 3 AE	NIV: 25.4%	PEM: 31.6%	D + T: 41%
	IPI: 55.2%	PLA: 18.5%	PLA: 14%
Most common AE	Skin reactions: 44.5%	All immune-related AE: 37.3%	Pyrexia: 63%
	Fatigue: 34.5%	Fatigue or asthenia: 37.1%	Fatigue: 47%
	Gastrointestinal: 25.2%	Skin reactions: 28.3%	Nausea: 40%
AE leading to discontinuation	NIV: 9.7%	PEM: 13.8%	D + T: 26%
	IPI: 42.6%	PLA: 2.2%	PLA: 3%

Abbreviations: AE: adverse event, D + T: dabrafenib plus trametinib, ECOG PS: Eastern Cooperative Oncology Group Performance Status, NIV: nivolumab, PEM: pembrolizumab: PLA: placebo, 95% CI: 95% confidence interval. * did not meet the prespecified interim analysis boundary of *p* = 0.000019.

**Table 2 cancers-14-05426-t002:** Patient characteristics.

Characteristics	Nivolumab (*n* = 54)	Pembrolizumab (*n* = 48)	Dabrafenib Plus Trametinib (*n* = 20)	All Patients (*n* = 122)
Median age (range)—year	61 (21–84)	57.5 (23–80)	57.5 (31–73)	59 (21–84)
Sex, no. (%)				
Male	35 (64.8)	31 (64.6)	9 (45.0)	75 (61.5)
Female	19 (35.2)	17 (35.4)	11 (55.0)	47 (38.5)
ECOG performance status, no. (%)				
0	32 (59.3)	24 (50.0)	9 (45.0)	65 (53.3)
1	7 (13.0)	5 (10.4)	1 (5.0)	13 (10.7)
Unknown	15 (27.8)	18 (37.5)	10 (50.0)	43 (35.2)
Disease stage, no. (%) AJCC 7				
III	34 (63.0)	48 (100)	20 (100)	102 (83.6)
Unspecified	3 (5.6)	1 (2.1)	1 (5.0)	5 (4.1)
IIIa	1 (1.9)	20 (41.7)	5 (25.0)	26 (21.3)
IIIb	10 (18.5)	19 (39.6)	9 (45.0)	38 (31.1)
IIIc	20 (37.0)	8 (16.7)	5 (25.0)	33 (27.0)
Resected IV	20 (37.0)	-	-	20 (16.4)
Primary tumor location, no. (%)				
Head or neck	6 (11.1)	6 (12.5)	2 (10.0)	14 (11.5)
Body	19 (35.2)	22 (45.8)	8 (40.0)	49 (40.2)
Extremities	17 (31.5)	13 (27.1)	6 (30.0)	36 (29.5)
Acral	1 (1.9)	3 (6.3)	1 (5.0)	5 (4.1)
Mucosal	1 (1.9)	-	-	1 (0.8)
Unknown primary	9 (16.7)	2 (4.2)	3 (15.0)	14 (11.5)
Unknown	1 (1.9)	2 (4.2)	-	3 (2.5)
Subtype, no. (%)				
Superficial spreading	16 (29.6)	25 (52.1)	11 (55.0)	52 (42.6)
Nodular	9 (16.7)	8 (16.7)	5 (25.0)	22 (18.0)
Acral lentiginous	-	2 (4.2)	1 (5.0)	3 (2.5)
Spindle cell	-	3 (6.3)	-	3 (2.5)
Unclear	29 (53.7)	10 (20.8)	3 (15.0)	42 (34.4)
LDH above ULN, no. (%)				
Yes	5 (9.3)	-	-	5 (4.1)
No	43 (79.6)	44 (91.7)	20 (100.0)	107 (87.7)
Unknown	6 (11.1)	4 (8.3)	-	10 (8.2)
Ulceration, no. (%)				
Yes	-	15 (31.3)	1 (5.0)	16 (13.1)
No	14 (25.0)	25 (52.1)	5 (25.0)	44 (36.1)
Unknown	40 (74.1)	8 (16.7)	14 (70.0)	62 (50.8)
BRAF mutation, no. (%)				
Yes	28 (51.9)	17 (35.4)	20 (100.0)	65 (53.3)
No	21 (38.9)	20 (41.7)	-	41 (33.6)
Unknown	5 (9.3)	11 (22.9)	-	16 (13.1)
NRAS mutation, no. (%)				
Yes	18 (33.3)	12 (25.0)	1 (5.0)	31 (25.4)
No	28 (51.9)	13 (27.1)	15 (75.0)	56 (45.9)
Unknown	8 (14.8)	23 (47.9)	4 (20.0)	35 (28.7)
KIT mutation, no. (%)				
Yes	-	1 (2.1)	-	1 (0.8)
No	45 (83.3)	31 (64.6)	17 (85.0)	93 (76.2)
Unknown	9 (16.7)	16 (33.3)	3 (15.0)	28 (23.0)
Previous systemic therapy for melanoma, no. (%)				
Adjuvant	1 (1.9)	1 (2.1)	1 (5.0)	3 (2.5)
Neo-adjuvant	1 (1.9)	-	3 (15.0)	4 (3.3)

**Table 3 cancers-14-05426-t003:** Adverse events.

Adverse Events, No. (%)	Nivolumab		Pembrolizumab		Dabrafenib Plus Trametinib	
	(*n* = 54)		(*n* = 48)		(*n* = 20)	
	Grade 1–2	Grade 3–4	Grade 1–2	Grade 3–4	Grade 1–2	Grade 3–4
Abdominal pain	2 (3.7)	-	4 (8.3)	-	1 (5.0)	-
Arthralgia	2 (3.7)	-	2 (4.2)	-	2 (10.0)	-
Chills	2 (3.7)	-	-	-	9 (45.0)	-
Cough	12 (22.2)	-	7 (14.6)	-	4 (20.0)	-
Diarrhea	22 (42.6)	1 (1.9)	9 (18.8)	2 (4.2)	1 (5.0)	-
Dyspnea	6 (11.1)	-	4 (8.3)	-	-	-
Fatigue or asthenia	43 (79.6)	-		-	16 (80.0)	-
Fatigue	42 (77.8)	-	36 (75.0)	-	16 (80.0)	-
Asthenia	1 (1.9)	1 (1.9)	1 (2.1)	-	1 (5.0)	-
Headache	16 (29.6)	-	8 (16.7)	-	10 (50.0)	-
Nausea, incl. vomiting	13 (24.1)	-	9 (18.8)	-	10 (50.0)	1 (5.0)
Pyrexia	15 (27.8)	-	4 (8.3)	-	13 (65.0)	-
Skin reaction	17 (31.5)	-	9 (18.8)	-	6 (30.0)	-
Pruritus	13 (24.1)	-	9 18.8)	-	3 (15.0)	-
Rash	9 (16.7)	-	2 (4.2)	-	5 (25.0)	-
Immune-related adverse events						
Colitis	3 (5.6)	-	1 (2.1)	1 (2.1)	-	-
Diabetes Mellitus type 1	-	-	-	-	-	-
Hepatitis	-	2 (3.7)	-	-	-	-
Hyperthyroidism	8 (14.8)	-	6 (12.5)	-	-	-
Hypothyroidism	11 (20.4)	-	9 (18.8)	-	-	-
Pneumonitis	-	-	2 (4.2)	-	-	-

**Table 4 cancers-14-05426-t004:** Treatment-limiting adverse events in patients who ended treatment due to toxicity. Patients could have one, or combinations of adverse events.

Adverse Events, No.	Nivolumab *n* = 9	Pembrolizumab *n* = 3	Dabrafenib Plus Trametinib *n* = 8
Allergic reaction	-	-	1
Arthalgia	1	1	1
Chills	-	-	3
Decreased appetite	-	-	1
Fatigue	-	-	2
Fever	-	-	5
Headache	-	-	1
Liver function disorders	-	-	1
Malaise	-	-	2
Myalgia	-	1	2
Nausea	-	-	3
Skin disorder	-	-	4
Syncope	-	-	1
Tachycardia	-	-	1
Immune-related adverse events	8	2	-
Adrenalitis	1	-	-
Colitis	2	1	-
Hepatitis	2	-	-
Hypocortisolism	1	-	-
Meningitis	1	-	-
Myocarditis, no.	1	-	-
Myositis, no.	1	-	-
Polymyalgia rheumatica, no.	1	-	-
Pneumonitis, no.	2	1	-
Thyroiditis, no.	2	-	-

## Data Availability

The data that support the findings of this study are available from the corresponding author, J.Z., upon reasonable request.

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
