# Peer review of "Application of Electronic Health Record Text Mining: Real-World Tolerability, Safety, and Efficacy of Adjuvant Melanoma Treatments"

_cancers, 2022, doi:10.3390/cancers14215426_

Round 1
Reviewer 1 Report
The Authors conducted a very interesting study on tolerability, safety, and efficacy of adjuvant melanoma immunotherapy using a text-mining approach.
This approach is very interesting and - in my opinion - the most fascinating element of the study. I understand that Cancers is mainly a clinical journal, but can the Authors expand on the text-mining approach they used and further comment on its strenghts and limitations in the Discussion?
Very minor, in the Introduction:
- the first paragraph can likely be omitted
- lines 71-83, are all values necessary in the Introduction?
Author Response
Reviewer 1:
The Authors conducted a very interesting study on tolerability, safety, and efficacy of adjuvant melanoma immunotherapy using a text-mining approach.
This approach is very interesting and - in my opinion - the most fascinating element of the study.
I understand that Cancers is mainly a clinical journal, but can the Authors expand on the text-mining approach they used and further comment on its strengths and limitations in the Discussion?
Dear reviewer, thank you for your comments. We have further extended the strengths and limitations section in the Discussion as follows:
Line 347 – 358:
“We used rule-based text-mining to detect patients, and collect characteristics, outcomes, and adverse events from electronic health records. The advantage of this method is the faster and more standardized data extraction. Furthermore, the set of queries can repeatedly be reused to review the status of adjuvant melanoma treatments in this hospital in the future, for example on yearly basis, and has the potential to be implemented in other hospitals treating melanoma patients.
However, due to its retrospective design, the extracted data is both limited by the information stored in the EHR, which is prone to have missing data [38], and by the terms included in the queries [39]. In this study we aimed for a high sensitivity, by using extensive lists of keywords for the searches in unstructured text, and additionally performed visual context analysis to confirm outcomes, resulting in limited missing data. However, certain information can be underreported.”
Very minor, in the Introduction:
- the first paragraph can likely be omitted
We agree that the first paragraph is somewhat extensive, therefore we have shortened it as shown below. However, we believe it is relevant to put the use of treatments in clinical perspective.
Line 46 – 51:
“In 2020, approximately 300 000 patients worldwide were diagnosed with melanoma of the skin, accounting for 1.7% of all cancer diagnoses [1]. In the Netherlands, the incidence of melanoma has more than doubled in the last 20 years from 18/100 000 people in 2001 to 43/100 000 in 2021 with a mortality rate of 4-5/100 000 [3].”
- lines 71-83, are all values necessary in the Introduction?
We agree that not all values are necessary. We have removed the incidence specified per adverse events since the relevant values are also included in the discussion.
Line 71 - 83:
“Grade 3 or 4 adverse events (AEs) occurred in 25.4% and 31.6% of the study populations of nivolumab and pembrolizumab versus 41% in dabrafenib plus trametinib treated patients, and 9.7% and 13.8% of the patients discontinued treatment due to toxicity, com-pared to 26% in the dabrafenib and trametinib treated. However, most reported AEs on ICI were skin reactions and fatigue. All immune-related (ir)AEs combined also have a high incidence, of which hypothyroidism and hyperthyroidism - both manifestations of thyroiditis - were most frequent, but also include, e.g., diabetes mellitus type I [7, 9]. Of these immune-related endocrine toxicities it is presumed they result in permanent and irreversible dysfunction, resulting in lifelong hormone supplementation [12]. This contrasts with the AEs of dabrafenib plus trametinib, of which pyrexia, fatigue and nausea are most common and easily reversible after treatment interruption [10, 13].”

Reviewer 2 Report
Reviewer comments and suggestions
The study highlights the application of text-mining which is a valuable method to collect data for the evaluation of adjuvant melanoma treatments. The author aimed to retrospectively review the benefits and risks of these treatments in clinical practice, by extracting data from electronic health records with a text-mining tool.
In the population of 122 patients, 55 used nivolumab, 48 used pembrolizumab, and 20 used D+T, and the study found that the ICI were better tolerated than D+T. The efficacy results including the recurrence-free survival were promising, however, it was a short follow up study
The paper was nicely written, and a few minor comments needed to consider before publication.
A few concerns/comments needed to be explained/modified.
- Line 91-92 Please add more evidence for validating text data mining for evaluating treatment or other study objectives in cancer.
- There should be some number for approval from Medical Ethics Review also did the authors take permission/ consent forms from the patients to utilize their data
- Figure 4 it would be nice if the author presents a few points based on the observation in the legend part
- Is section 4.4. Eligibility criteria are important to be mentioned in the discussion part
- All references need to be changed based on MDPI journal guidelines
Author Response
Please, see attachment.
